# Influence of Atmospheric Flow Structure on Optical Turbulence Characteristics

**Artem Y. Shikhovtsev** [1,*], **Pavel G. Kovadlo** [1], **Anatoly A. Lezhenin** [2], **Oleg A. Korobov** [3], **Alexander V. Kiselev** [1], **Ivan V. Russkikh** [1], **Dmitrii Y. Kolobov** [1] **and Maxim Y. Shikhovtsev** [1]

1 Institute of Solar-Terrestrial Physics, The Siberian Branch of the Russian Academy of Sciences, Irkutsk 664033, Russia

2 Institute of Computational Mathematics and Mathematical Geophysics, The Siberian Branch of the Russian Academy of Sciences, Novosibirsk 630090, Russia

3 Faculty of Information Technologies, Novosibrsk State University, Novosibirsk 630090, Russia

* Correspondence: ashikhovtsev@iszf.irk.ru; Tel.: +7-908-6464257

**Abstract:** This article discusses the quality of astronomical images under conditions of moderate small-scale turbulence and varying meso-scale airflows above the Baikal Astrophysical Observatory (BAO). We applied a Weather Research and Forecasting (WRF) Model, as well as statistical estimations of the Fried parameter from the differential motion of the solar images. The simulations were performed with a fairly high horizontal resolution within a large area of $1600 \times 1600$ km. A high horizontal resolution provides representative estimations of atmospheric characteristics and correct accounting of large-scale air advection. We considered the influence of atmospheric motions over the cold water area of Lake Baikal, as well as meso-scale vortex structures over rough terrain on solar image quality. A better understanding of structured turbulent small-scale motions and optical turbulence over rough terrain may help to develop advanced methods for diagnostics and prediction of image quality. For the first time, we have shown that the BAO is located at the periphery of a meso-scale atmospheric vortex structure with an anticyclonic direction of airflows in the daytime. An increase in image quality was associated with weakening airflows over Lake Baikal and a decrease in the intensity of wind speed fluctuations. Calculated spectra of atmospheric turbulence in the daytime were close to the classical form. At night and in the morning, the spectra had a steeper slope on small scales. Deformations of the spectra were due to the suppression of turbulence under stable stratification of the atmosphere. The characteristic horizontal scales of the transition from "$-5/3$" to $\sim$"$-3$" spectral slope were 2–2.5 km. The results obtained using the WRF model and analysis of optical turbulence strength (namely, the Fried parameter) indicated that the parameterization schemes used in the WRF model were accurate.

**Keywords:** turbulence; meso-scales; WRF

## 1. Introduction

The structure of small-scale turbulence significantly depends on the orography and energy of meso-scale and large-scale atmospheric flows. In particular, energy and enstrophy transfer in a spectrum of atmospheric flows determine the conditions of small-scale turbulence evolution [1,2]. A number of numerical and experimental studies have focused on the influence of meso-scale and large-scale airflows on small-scale turbulence [3–5]. Over the years, meso-scale models have been used to obtain optical turbulence parameters above sites of interest for astronomy [6,7]. A powerful means to simulate atmospheric processes is a Weather Research and Forecasting (WRF) model, including partial models to diagnose and forecast the weather [8–14]. In [15], the authors studied turbulence formed in jet streams. Jayaraman B. et al. identified the influence of residual turbulence from outside the jet on the intensity of turbulent fluctuations inside the jet. However, it can be difficult to unambiguously and reliably assess the relationship between turbulence inside and outside

the jet through simulation. This is due to both the limitations that arise when matching the schemes of parameterization of meso-scale and micro-scale motions, and when motions in the central part of the jet has weakened interactions with motions outside the jet.

Interesting studies on the origin and evolution of turbulence have been devoted to the phenomenon of coherent and structured optical turbulence. The phenomenon of coherent turbulence leads to a weakening of the turbulent phase fluctuations of a light wave propagating in the stratosphere–troposphere. The attenuation of phase fluctuations has been confirmed by a number of observations of image motion caused by fluctuations in the air refractive index along the line of sight of ground-based solar telescopes [16,17].

Wasson G. et al. simulated meteorological conditions under which clear air turbulence was generated over northern India. The authors used the Weather Research and Forecasting model [18]. In the WRF model, a revised MM5 Monin–Obukhov scheme and Yonsei University scheme (YSU) were used to describe the surface and atmospheric boundary layers. Calculations were carried out for the two domains with a spatial resolution of 6 and 2 km, respectively. The authors showed that the WRF model could reproduce upward and downward airflows with high accuracy and describe the characteristics of clear air turbulence.

Based on optical turbulence theory, we need to describe optical distortions using scales ranging from a few centimeters to so-called outer scales of turbulence (tens of meters). However, the WRF model does not reproduce such spatial scales. Eddy resolution models are also significantly limited, especially for long time intervals. Simulation complexity is associated with incorrect assessments of the influence of large-scale and meso-scale disturbances on small-scale turbulence.

C. Giordano et al. pointed out the ability of the WRF model to reproduce weather and optical conditions, even with low-resolution input data [19]. Diaz-Fernandez et al. [20] state that spatial scales ∼3 km should be revealed in order to simulate atmospheric processes and evaluate their influence on small-scale turbulence. For example, mountain lee waves influencing small-scale turbulence may be detected in these simulations. Comparing the WRF and HARMONIE models (with a spatial resolution of 1 km), the authors showed that the turbulence generated in mountain lee wave events was reproduced in 86 and 55% of cases, respectively. We also note that the authors used the Yonsei University scheme to parameterize the atmospheric boundary layer.

In addition to diagnosing atmospheric situations, the WRF model is beginning to be successfully used to forecast atmospheric characteristics, including parameters of surface and high-altitude turbulence. The strength of small-scale turbulence is usually estimated by $C_n^2$. The structure constant of air refractive index fluctuations, $C_n^2$, is associated with turbulent fluctuations in air temperature (or air density). In particular, one study used the WRF model to forecast $C_n^2$ in the surface layer [21]. Yang Q. et al. showed that the WRF model reproduces the measured $C_n^2$ with good accuracy. The highest deviations between the modeled and measured values were observed under weak winds and in the presence of temperature inversions. The WRF model has also been used to describe optical turbulence in applications to ground-based astronomical telescopes for the Tibetan Plateau [22]. The authors showed that meso-scale models could provide reliable estimations of optical turbulence parameters in rough terrain conditions above an astronomical site on the Tibetan Plateau. The major aim of the present study was to identify possible ways in which large-scale and meso-scale atmospheric structures influence small-scale turbulence and image quality within the observatory region. In order to diagnose and predict small-scale turbulence and solar image quality, the present study was aimed at expanding knowledge about structured turbulent small-scale motions and optical turbulence.

In this article, we compare measurements of solar image differential motion with WRF model data. The relevance of using the WRF model to describe the atmospheric states over observatories was determined by:



(i)     The correct accounting of large-scale advections and diurnal transformations of air masses in domains with relatively high horizontal resolution.
(ii)    Reproducibility of meso-scale structures affecting small-scale turbulence.
(iii)   The reconstruction of vertical profiles of air temperatures and wind speeds with high temporal and vertical resolution. Using gradient approaches, these data make it possible to estimate the vertical profiles of optical turbulence, as well as the Fried parameter and seeing.
(iv)    The clarification or creation of special parameterization schemes that relate the characteristics of meso-scale atmospheric disturbances and small-scale turbulence [23,24]. These schemes can be used for reliable estimations of the wind speed, air temperature and refractive index fluctuations.
(v)     Clarification of spatial locations with the best astroclimatic parameters.

This study focuses on the summer season. In Section 2, we describe the configuration of the Weather Research and Forecasting model, as well as a method to estimate the strength of optical turbulence. Section 3 is devoted to a comparison of the Fried parameter with wind speed characteristics deduced from WRF simulations. In Section 4, we discuss the results of our calculations. Our final conclusions are drawn in Section 5.

## 2. Used Methods

### 2.1. Adaptation of the Weather Research and Forecasting Model within the Baikal Astrophysical Observatory Region

Astronomical observations require high-quality diagnostics and prediction of the main atmospheric parameters, which determine the image quality. These parameters include wind speed components, air temperature, the refractive index at different heights in the atmosphere, as well as the Fried parameter and seeing.

In order to study the influence of large-scale and meso-scale flows on small-scale turbulence, we chose the region where the Baikal Astrophysical Observatory (BAO) and Sayan Solar Observatory (SSO) are located. In this paper, we focus our attention on the Baikal Astrophysical Observatory. This observatory is located in Listvyanka, 70 km from the city of Irkutsk. The main instrument in the observatory is the large solar vacuum telescope, which is located on a hilltop near Lake Baikal. The atmospheric circulation over the observatory has a complex character. The large-scale air flow interacts with the local mountain–valley air circulation. Episodic observations show that meso-scale jet streams are often observed above the observatory within the atmospheric boundary layer. For simulation of atmospheric flows over the observatory and Lake Baikal, we used the non-hydrostatic WRF model (version 3.9.1). The WRF model consists of a Eulerian mass solver with fully compressible non-hydrostatic equations, terrain following vertical coordinates and a staggered horizontal grid with complete Coriolis and curvature terms [25].

All the simulations were performed with the following constant 3D domains:

(i)     The largest domain had a coarse grid with a horizontal resolution of 8 km over a 1600 × 1600 km area. The largest domain covered Lake Baikal, the Eastern Sayan mountains and surrounding areas, and included radio sounding stations (Angarsk and Nizhneudinsk).
(ii)    The nested domain with a ratio of 1/4 corresponded to a 400 × 400 km area. The horizontal resolution in the domain was 2 km.
(iii)   The smallest domain above the BAO had a fine horizontal resolution of 500 m. The area of interest was limited to 100 × 100 km.
(iv)    There were 44 vertical levels, with higher resolution within the lower layers of the atmosphere. The number of height levels, up to 3100 m, was 12 (Figure 1). In higher layers of the atmosphere, the simulation was performed up to 30,800 m.

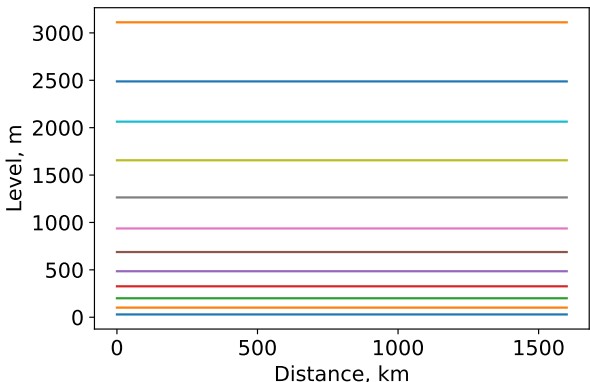

**Figure 1.** The number of height levels, up to 3100 m.

Figure 2 shows three domain configurations for the WRF simulation. These domains are centred on the BAO. In comparison with Mount Graham (110 km northeast of Tucson, Arizona, United States), the site of the large binocular telescope, we performed a simulation within a larger area (1600 × 1600 km). The outermost model size used by Hagelin S. et al. was 800 × 800 km [26]. In addition, Wang H. et al. used three nested domains in WRF simulations to search for the best site in the Ali area [27]. However, calculations were performed on a grid with lower horizontal resolutions (25, 5 and 1 km, respectively). Yang Q. et al. used the same horizontal resolutions for estimating astronomical seeing above Dome A using the Polar WRF model [23].

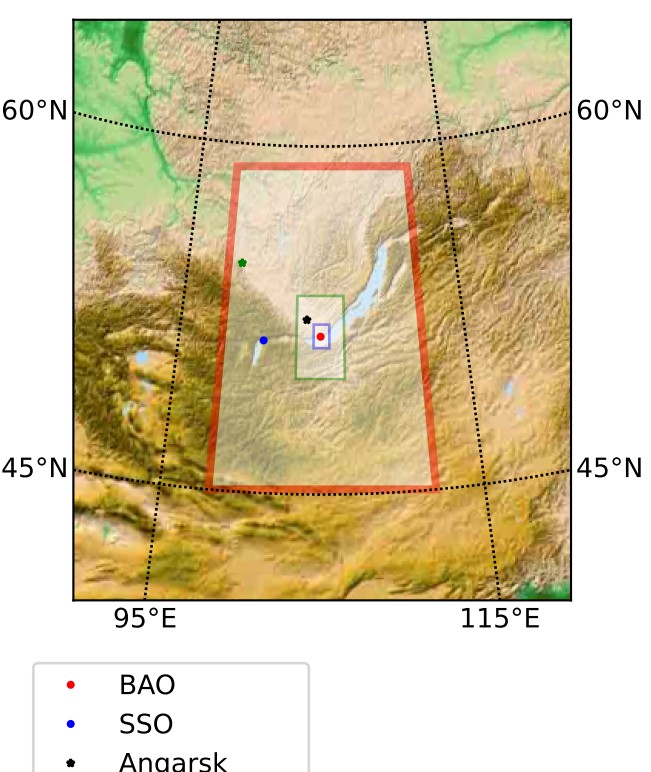

**Figure 2.** Three domain configurations for WRF simulation. The smallest domain covers the BAO, shown by the red marker, with 500 m horizontal grid resolution. The intermediate-sized domain covers the radiosounding station Angarsk. The largest domain is shown by the red rectangle. The Sayan Solar Observatory (SSO) and radio sounding station Nizhneudinsk are located in this domain.

As the initial and boundary conditions, we used NCEP Global Forecast System (GFS) data. These initial data have a $1 \times 1°$ horizontal resolution. The data assimilation system uses the maximum number of satellite and meteorological observations. Data assimilation and forecasts were made four times a day, at 00, 06, 12 and 18 UTC. Based on meteorological data and topography within the BAO region, we used parameterization schemes for the configuration of the WRF model, as shown in Table 1.

**Table 1.** Parameterization schemes in the WRF model.

| Physical Schemes of Parametrization | Description |
| --- | --- |
| Yonsei University scheme | Atmospheric boundary layer |
| MM5-similarity scheme | Surface layer |
| Kain–Fritsch scheme | Cloudiness |
| Simple scheme based on Dudhia | Short-wave radiation |
| Rapid Radiative Transfer Model scheme | Long-wave radiation |
| Thompson scheme | Microphysical processes in clouds |
| RUC scheme | Land surface model |

We should note that the WRF model separately parameterizes the processes in the atmospheric boundary layer and the so-called surface layer. The surface layer exhibits the most dramatic temporal and height variations in mean meteorological characteristics and turbulent parameters, including $C_n^2$ [28,29]. The choice of the Yonsei University parameterization scheme for the BAO and SSO region was confirmed by recent studies on atmospheric processes [30]. Within the middle- and high-latitude regions of China, as well as the Baikal region, various parameterization schemes were investigated, including the Medium-Range Forecast scheme (MRF), the Mellor–Yamada Nakanishi and Niino Level 2.5 scheme (MYN), the Bougeault and Lacarrere scheme (BLS), the Yonsei University scheme (YSU), the Asymmetric Convective Model Version 2 scheme (ACM), the Grenier–Bretherton–McCaa scheme (GBM) and the University of Washington moist turbulence scheme (UWS). Ma H. et al. estimated the performances of these parameterization schemes and the recovery accuracy of wind speeds at heights of 10 and 100 m, the surface air temperature at 2 m and surface atmospheric pressure. The largest correlation coefficients and smallest root mean square deviations between the measured and simulated wind speeds (as well as air temperatures) corresponded with the MRF and YSU schemes.

In our study, we performed about 240 h of simulations above the BAO and SSO. Due to the need for diagnostics and forecasting of atmospheric characteristics that determine the quality of solar images, we selected only daytime hours.

In the WRF model, we applied a combination of the YSU and revised MM5 parametrization schemes. Vertical diffusion of parameter $C$ is described by the equation:

$$\frac{\partial C}{\partial t} = \frac{\partial}{\partial z}\left(K\left(\frac{\partial C}{\partial z} - \gamma\right) - \overline{w'c'}\left(\frac{z}{h_{ABL}}\right)^3\right), \tag{1}$$

where $K$ is the turbulent diffusion coefficient, $z$ is the height, $h_{ABL}$ is the height of the atmospheric boundary layer (ABL) and $t$ is time. The YSU scheme determines the ABL height to be where the bulk Richardson number exceeds a threshold value of 0. The first term on the right-hand side of Equation (1) describes vertical nonlocal air mixing by convective eddies. $\gamma$ is a counter-gradient transport term that incorporates the contribution of convective eddies to the total flux. $\overline{w'c'}$ is the vertical turbulent flux at the inversion layer.

The turbulent diffusion coefficient is calculated from:

$$K = kw_s z\left(1 - \frac{z}{h_{ABL}}\right)^p, \tag{2}$$

$$w_s = u_*/\phi_m, \tag{3}$$

where $p = 2$, $k$ is the von Karman constant, $w_s$ is the mixed-layer velocity scale, $u_*$ is the friction velocity and $\phi_m$ is the non-dimensional function described by Hu X-M. et al. [31]. This model uses a parabolic $K$-profile in an unstable mixed layer with the addition of an explicit term to treat the entrainment layer at the top of the ABL.

*2.2. Method to Estimate the Strength of Small-Scale (Optical) Turbulence*

To calculate the main characteristic of image quality, namely the Fried parameter, we used the classical method based on relation between differential image motion and the Fried parameter [32]. The formula to calculate the Fried parameter $r_0$ can be written as follows:

$$\sigma_\alpha^2 = K\lambda^2 r_0^{-5/3} D^{-1/3}, \tag{4}$$

where $\lambda$ is the light wavelength and $D$ is the telescope diameter. In Formula (4), the parameter $r_0$ is considered to be a function of the light wavelength and the structure constant of air refractive index fluctuations ($C_n^2$). The coefficient $K$ used in Formula (4) depends on the ratio of the distance between the centers of the subapertures $S_d$ and the subaperture diameter $d_s$, as well as the direction of image motion and type of wavefront slope. In the present study, we used the formulas for longitudinal and transverse coefficients:

$$K_l = 0.34 \left( 1 - 0.57 \left( \frac{S_d}{d_s} \right)^{-1/3} - 0.04 \left( \frac{S_d}{d_s} \right)^{-7/3} \right), \tag{5}$$

$$K_t = 0.34 \left( 1 - 0.855 \left( \frac{S_d}{d_s} \right)^{-1/3} + 0.03 \left( \frac{S_d}{d_s} \right)^{-7/3} \right). \tag{6}$$

The measurements of differential sunspot image motion were performed using the Shack–Hartmann wavefront sensor (SHWS). SHWS is the main component of the adaptive optics system of the large solar vacuum telescope [33].

## 3. Integral Strength of Optical Turbulence and Spatial Distributions of Surface Wind Speeds over the BAO

Using the observations of sunspot subimages to calculate the Fried parameter, we estimated the daytime changes of $r_0$. Figure 3 demonstrates daytime changes in the Fried parameter at the BAO site for 8 August 2022.

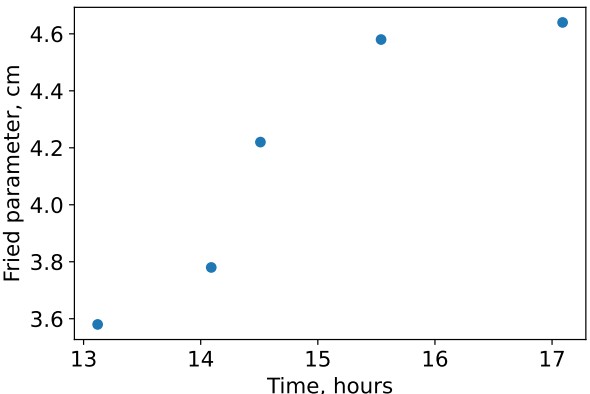

**Figure 3.** Daytime changes in the Fried parameter at the BAO site for 8 August 2022.

The analysis of daytime changes in the Fried parameter shows that sunspot image quality increases from noon to evening. This scenario does not reflect the usual changes in image quality during the day.

In order to improve knowledge about the formation of optical turbulence above the BAO and the influence of atmospheric meso-scale and large-scale movements on it, we calculated the spatial distributions of wind speeds at a height of 200 m above the

observatory. We should emphasize that the horizontal resolution of the obtained spatial distributions was 500 m, and the temporal resolution was 3 min. Figure 4a–c shows the spatial distributions of the wind speeds at a height of 200 m above the BAO within the third domain, under conditions of strong optical turbulence. Figure 5a,b shows the wind speed distributions under conditions of moderate intensity of optical turbulence. Analyzing Figures 4 and 5, we note that an increase in the Fried parameter is accompanied by changes in airflow structure in the lower layers of the atmosphere, which makes the greatest contribution to the quality of solar images. The lowest values of the Fried parameter correspond to strong airflows over Lake Baikal. An observed increase of the Fried parameter by 1 cm during the day can be associated with a general decrease in surface wind speeds over Lake Baikal. In the daytime, a meso-scale vortex structure with anticyclonic circulation of air flows forms over the BAO. The velocity of these flows in the vortex structure is low. According to our measurements, the velocity varied from 0.5 to 3 m/s. In order to identify features of the spatial structure of airflows, we also presented the spatial distributions of wind speeds within the third domain, both in the morning and at night (Figure 6a,b).

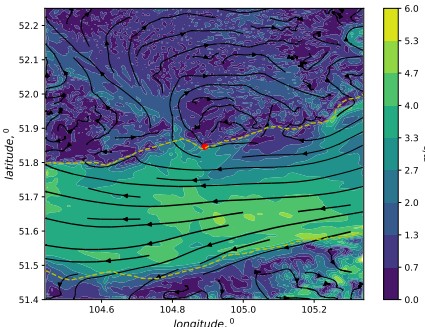

(**a**) 8 August 2022 (13 h 12 min), $r_0 = 3.58$ cm.

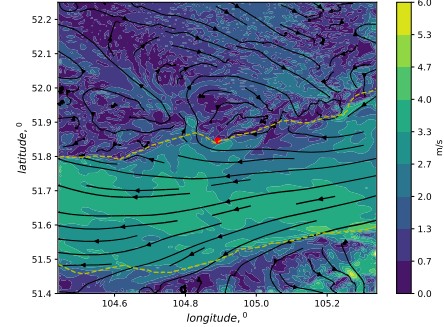

(**b**) 8 August 2022 (14 h 09 min), $r_0 = 3.78$ cm.

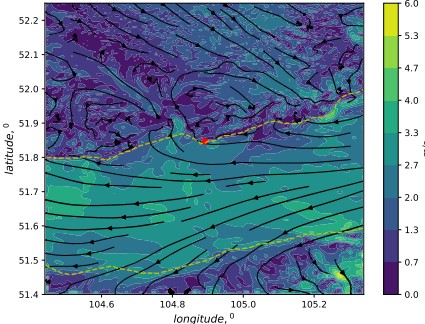

(**c**) 8 August 2022 (14 h 51 min), $r_0 = 4.22$ cm.

**Figure 4.** Spatial distributions of wind speeds in the third domain with a horizontal resolution of 500 m under conditions of strong optical turbulence. The streamlines are shown by arrows. The red marker corresponds to the BAO location. The yellow dotted line is the coastline of Lake Baikal.

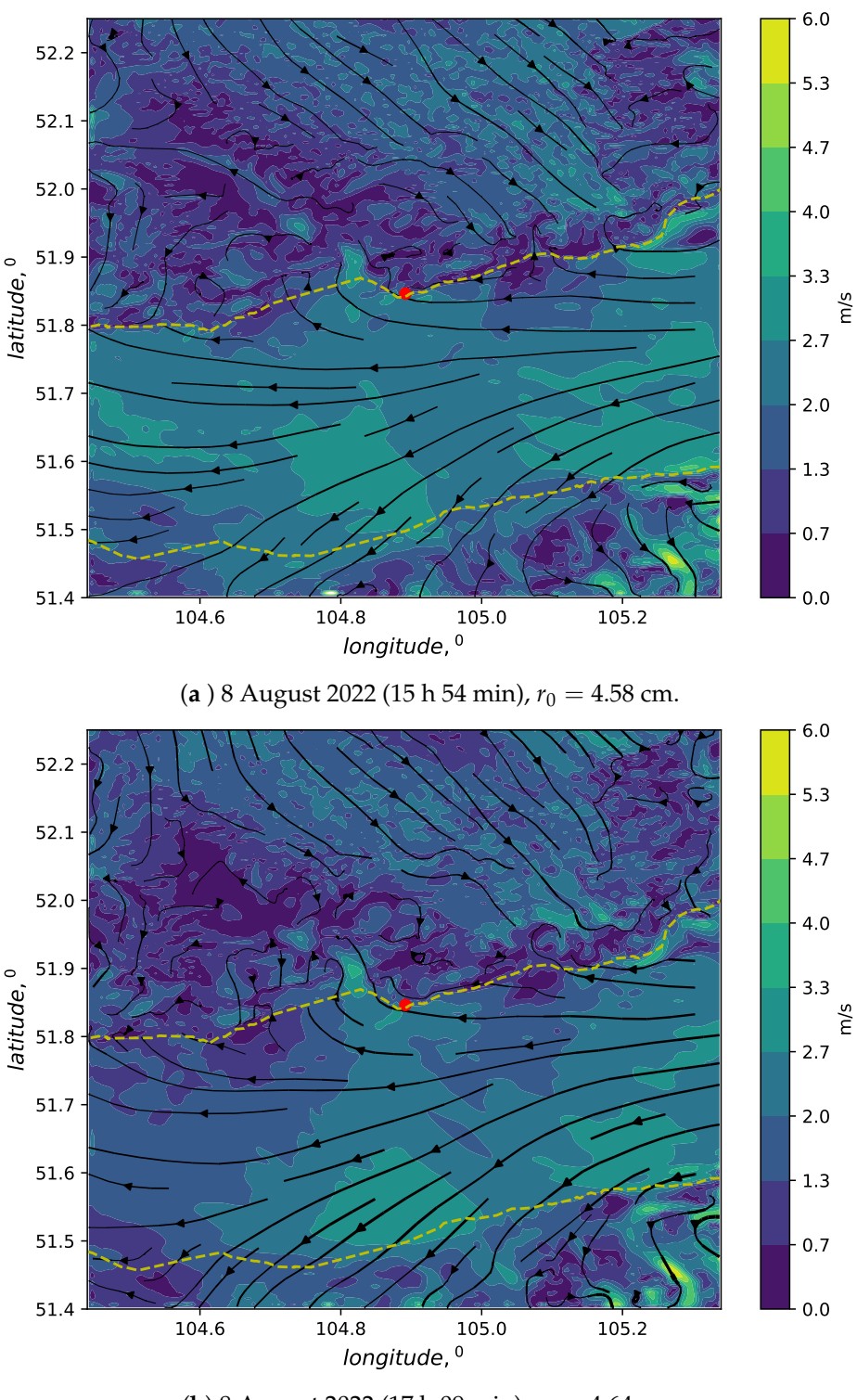

(**a**) 8 August 2022 (15 h 54 min), $r_0 = 4.58$ cm.

(**b**) 8 August 2022 (17 h 09 min), $r_0 = 4.64$ cm.

**Figure 5.** Spatial distributions of wind speeds under conditions of moderate intensity of optical turbulence. The streamlines are shown by arrows. The red marker corresponds to the BAO location. The yellow dotted line is the coastline of Lake Baikal.

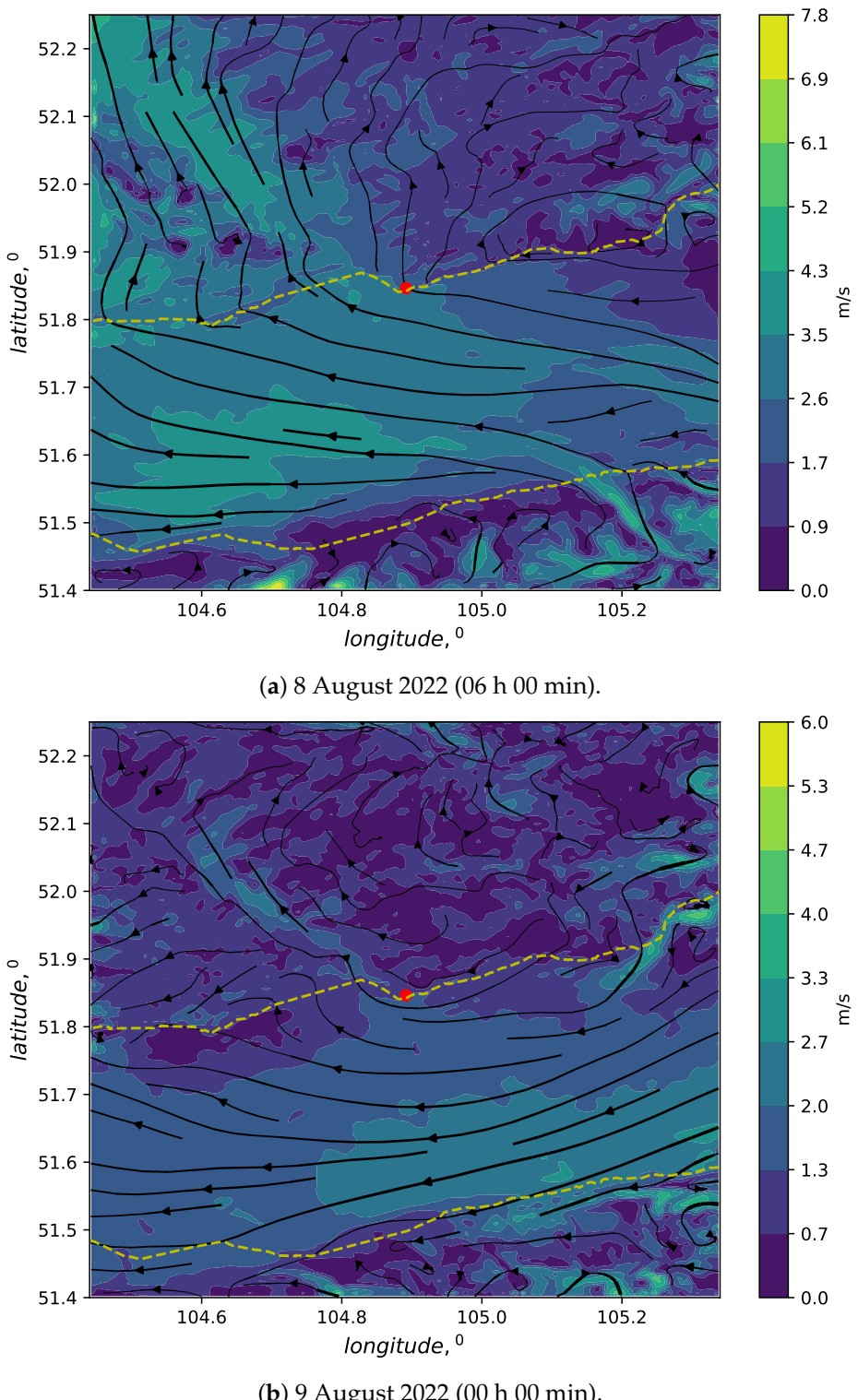

(**a**) 8 August 2022 (06 h 00 min).

(**b**) 9 August 2022 (00 h 00 min).

**Figure 6.** Spatial distributions of wind speeds in the third domain, with a horizontal resolution of 500 m, measured in the morning and at night. The streamlines are shown by arrows. The red marker corresponds to the BAO location. The yellow dotted line is the coastline of Lake Baikal.

*Analysis of Structure of Atmospheric Flows in the Lower Layers*

In order to determine the influence of meso-scale structures in the lower layers on small-scale and optical turbulence, we performed a spectral analysis of the wind speed. Figures 7–10 show the spatial changes of wind speeds along the meridian, passing through

the BAO (104.89166 E) (black lines). The blue lines correspond to spatial variations in the high-frequency components of wind speed fluctuations (spatial scales over 12 km were filtered). These figures also show the dimensionless energy spectra of wind speed fluctuations without filtering (blue lines in the lowest panels) and with low-frequency filtering (orange lines). The green lines correspond to the dependence of the power spectral density (PSD) of wind speed fluctuations on the scale to the power of "5/3". Statistics of the wind field are listed in Table 2. The second column of the table shows the wind speed averaged over Lake Baikal. The third and fourth columns show the spectral characteristics of wind speed fluctuations. In particular, the spectral characteristic $E_n$ is calculated by the formula:

$$E_n = \frac{\sigma}{\sigma_{max}},$$ (7)

where $\sigma$ is the mean square deviation of high-frequency fluctuations in wind speed determined over land and $\sigma_{max}$ is the maximum value of $\sigma$.

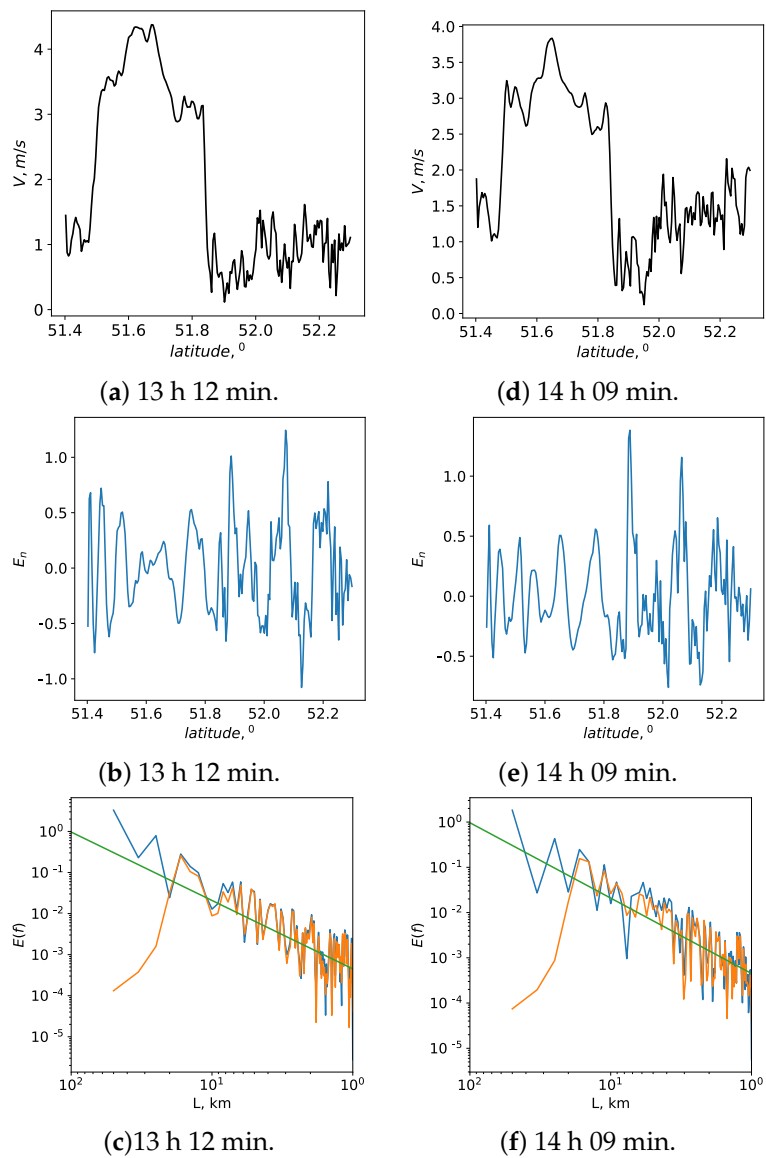

**Figure 7.** Distributions of atmospheric characteristics along meridian 104.89166 E at 13 h 12 min under low image quality conditions, including (**a**) wind speed and (**b**) high-frequency wind speed fluctuations. Subfigure (**c**) shows the energy spectra of spatial wind speed fluctuations calculated without filtering (blue lines) and with low-pass filtering (orange lines). Subfigures (**d**–**f**) show atmospheric characteristics as in figures (**a**–**c**), but at 14 h 09 min.

The parameter $E_n$ characterizes the spatial variability of high-frequency fluctuations in wind speed over land on a scale ranging from 500 m to 12 km. To identify features in the scales of wind speed fluctuations, we also calculated the integral of the power spectral density of wind speed fluctuations over the entire range of scales. Table 2 shows the integral normalized to its maximum value *Int*.

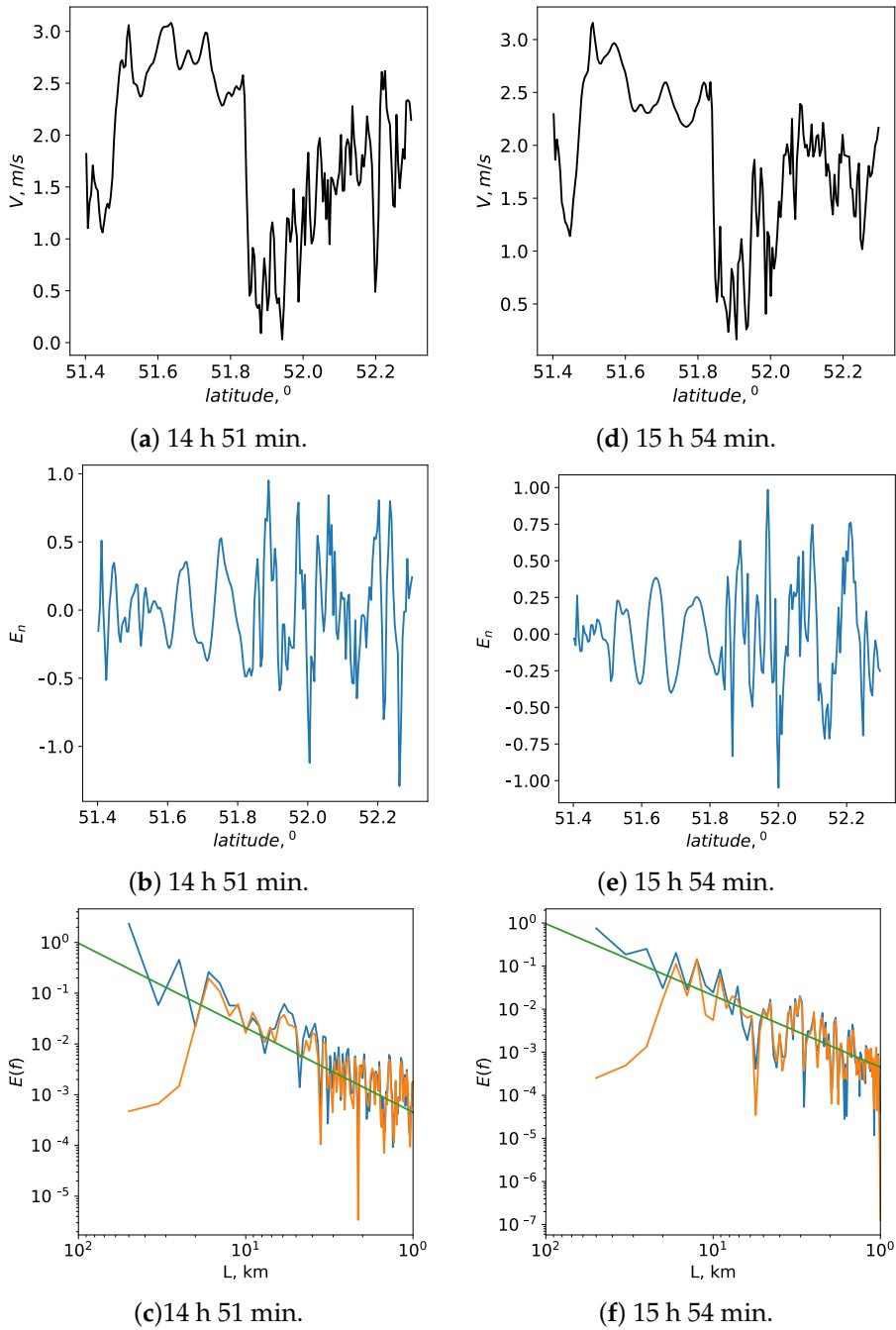

**Figure 8.** Distributions of atmospheric characteristics along meridian 104.89166 E at 14 h 51 min, including (**a**) wind speed and (**b**) high-frequency wind speed fluctuations. Subfigure (**c**) shows the energy spectra of spatial wind speed fluctuations calculated without filtering (blue lines) and with low-pass filtering (orange lines). Subfigures (**d**–**f**) show atmospheric characteristics as in figures (**a**–**c**), but at 15 h 54 min.

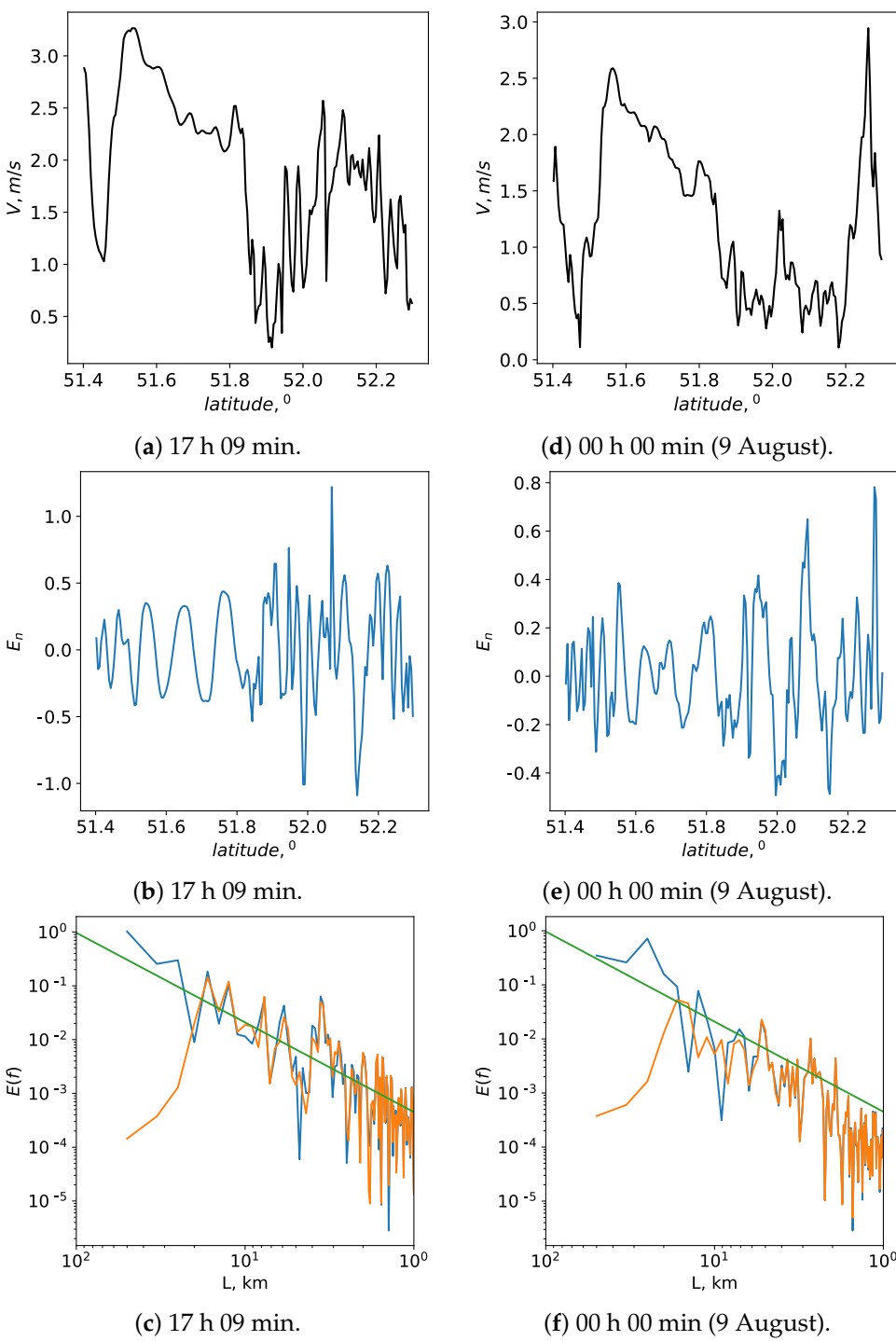

**Figure 9.** Distributions of atmospheric characteristics along meridian 104.89166 E in the evening (17 h 09 min), including (**a**) wind speed and (**b**) high-frequency wind speed fluctuations. Subfigure (**c**) shows the energy spectra of spatial wind speed fluctuations calculated without filtering (blue lines) and with low-pass filtering (orange lines). Subfigures (**d–f**) show atmospheric characteristics as in figures (**a–c**), but at night (00 h 00 min).

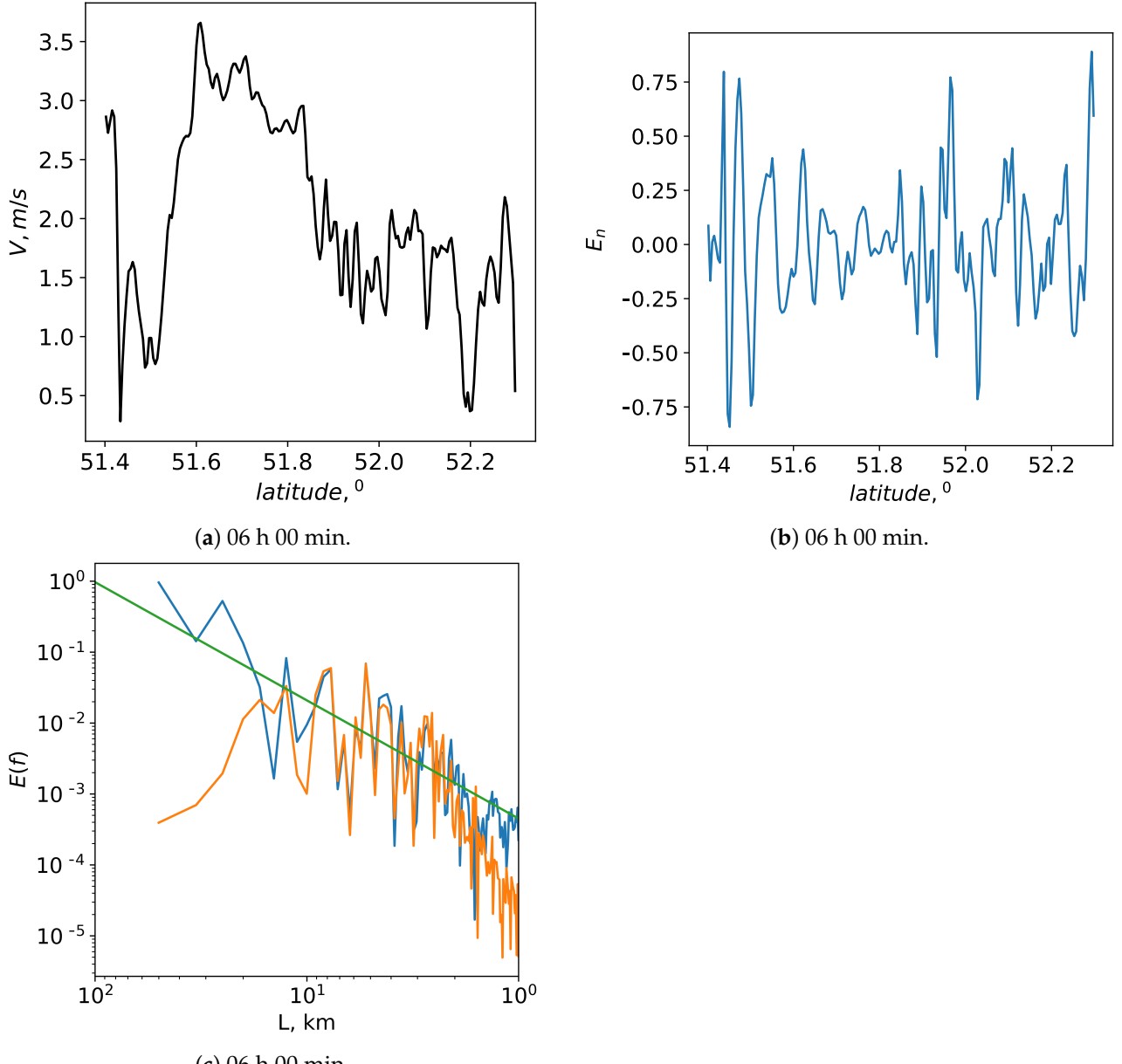

**Figure 10.** Characteristic distributions of atmospheric characteristics along the meridian 104.89166 E in the morning, including (**a**) wind speed and (**b**) high-frequency wind speed fluctuations. Subfigure (**c**) shows the energy spectra of spatial wind speed fluctuations calculated without filtering (blue lines) and with low-pass filtering (orange lines).

Spatially inhomogeneous fields of wind speeds were formed in the vicinity of the BAO. Despite the existence of a meso-scale vortex structure with an anticyclonic direction of airflows above the BAO, the intensity of optical turbulence along the line of sight was significant. In this case, the energy of atmospheric turbulence was predominantly determined by large-scale atmospheric processes, characterized by rather high wind speeds.

**Table 2.** Statistics of the wind field.

| Time | $V_m$, m/s | $E_n$ | $\int_{f_1}^{f_2} E(f)df/Int$ |
| --- | --- | --- | --- |
| 8 August | | | |
| 06 h 00 min | 2.7 | 0.38 | 0.51 |
| 13 h 12 min | 3.3 | 1.00 | 1.00 |
| 14 h 09 min | 2.9 | 0.93 | 0.85 |
| 14 h 51 min | 2.6 | 0.91 | 0.78 |
| 15 h 54 min | 2.3 | 0.77 | 0.65 |
| 17 h 09 min | 2.1 | 0.85 | 0.76 |
| 9 August | | | |
| 00 h 00 min | 1.7 | 0.34 | 0.29 |

Analyzing the changes in the Fried parameter, we can note that its minimum values were observed during intensive meso-scale turbulence both in the entire considered spectral range and in the range from 500 m to 12 km. Decrease in wind speed meso-scale fluctuations was especially pronounced in the morning and at night. Accordingly, the parameters $E_n$ and $\int_{f_1}^{f_2} E(f)df/Int$ were about 0.3–0.4 and 0.3–0.5, respectively. In the daytime, these parameters increased by 2.5–3.3 and 2.0–3.4 times, respectively. The daytime spectrum of atmospheric meso-scale turbulence was close to the classical spectrum of strong turbulence. In the night and morning, the spectrum had a steeper slope on small scales. We associate this spectral behaviour with a suppression of atmospheric turbulence on scales of 2–2.5 km in the surface layer, under conditions of stable thermal stratification. In these spectra, the positions of the peaks show that the characteristic horizontal scales of the meso-scale vortex structure with an anticyclonic direction of airflows, the center of which is formed north of the BAO, vary from 10 to 20 km.

## 4. Discussion

In the present article, we studied the relationship between variations in the strength of small-scale (optical) turbulence along the line of sight of a telescope and meso-scale atmospheric disturbances. The Fried parameter was chosen as a measure characterizing the strength of small-scale (optical) turbulence. As the Fried parameter increased, the strength of turbulence and its effect on the wavefront decreased. In order to identify meso-scale atmospheric disturbances, we used the WRF model with a resolution of 500 m in the third (internal) domain. This spatial resolution was chosen due to the fact that atmospheric fluctuations with a linear size of 500 m are smaller than a spatial scale corresponding to the micrometeorological maximum in the energy spectrum of atmospheric turbulence.

The WRF model was based on the use of the turbulent diffusion coefficients (for wind speed $K_V$ and air temperature $K_T$). We should note that the amplitudes and height profiles of diffusion coefficients can differ significantly under conditions of stable atmospheric stratification. These differences are associated with the features of the formation and suppression of turbulent fluctuations in wind speed and air temperature, as well as with the variations in the ratio between turbulent kinetic energy and turbulent potential energy. In general, the structure of small-scale turbulence in a stably stratified atmospheric boundary layer at night significantly differs from daytime turbulence. Measurement data show that daytime turbulence is more intense than nighttime turbulence. The energy spectrum of turbulent fluctuations is significantly deformed in stably stratified atmospheric layers. Figure 11 shows the dimensionless energy spectra of angle-of-arrival fluctuations caused by turbulence at a height of 2 m above surface. The low-frequency fluctuations in these spectra were suppressed.

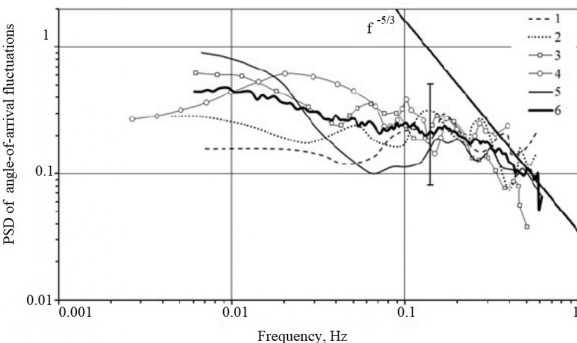

**Figure 11.** Energy spectra of angle-of-arrival fluctuations caused by turbulence within a stably stratified atmospheric surface layer. Lines 1–5 correspond to different measurement hours. Line 6 corresponds to the averaged spectrum

The WRF model provides a few schemes to describe the Earth's surface. In this study, we used the RUC scheme. This scheme solves equations of energy balance and moisture on the surface by considering six layers of soil (and phase changes of water in the soil during cold periods) [34]. S.A. Lysenko and P.O. Zaiko [35] showed that a high spatial resolution provides more accurate estimates of turbulent flows of latent and sensible heat from the underlying surface. We believe that a sufficiently high horizontal resolution would lead to accurate simulations of surface net shortwave radiation and reliable estimates of air temperature in the lower layers of the atmosphere.

Using the WRF model, the spatial distributions of wind speeds over the observatory, Lake Baikal and the surrounding region were obtained. Analysis of these distributions showed that a meso-scale vortex structure was formed above the observatory. The energy spectrum of motions within this vortex structure is described by the dependence of the power spectral density on frequency to a power close to minus 3. Such a spectrum is associated with the suppression of low-frequency turbulence.

In order to identify features in the vertical structure of wind speed, we obtain vertical wind speed profiles using the WRF model. Figure 12a,b shows vertical profiles of wind speeds over the observatory and Lake Baikal, respectively. The vertical wind speed profiles have a classical shape. However, an inter-comparison of these vertical profiles showed that a meso-scale jet stream was formed above Lake Baikal at a height of 1600 m. The wind speed on the axis of this jet stream varied from 8.8 to 10 m/s for the selected measurement periods. Over land, the jet stream was less pronounced.

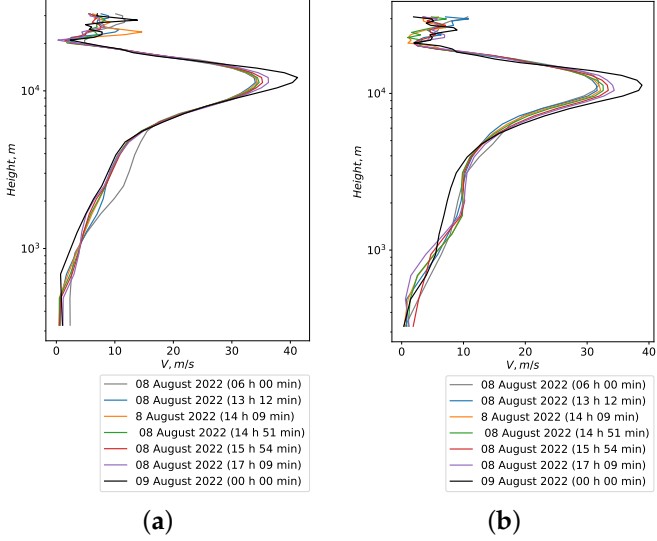

**Figure 12.** Vertical profiles of wind speed (**a**) over the observatory and (**b**) over Lake Baikal.

## 5. Conclusions

We performed a study of the atmospheric flows above the observatories and Lake Baikal. The WRF model accurately reproduced the airflows over the surface of the Lake, in the coastal area, as well as in the valley of River Angara. We should emphasize that this paper describes airflows in rough terrain with breeze air circulation, which significantly affects the vertical transfer of momentum and heat. The vertical transfer intensity depends on the large-scale air flow that suppresses vertical motions, as well as on the horizontal distributions of air temperature and local upward airflows. These physical representations were confirmed by a simulation of a boundary layer over mountainous terrain, described in [36].

Similar atmospheric situations were also simulated in [37]. Di Bernardino et al. used the WRF model to diagnose breeze circulation and the interaction of large-scale, meso-scale and micro-scale movements in summer.

Our results can be formulated as follows:

(i)   We adapted the WRF model for the Baikal Astrophysical Observatory and Sayan Solar Observatory region. We shown that the YSU parametrization scheme reproduced the local air circulation during the day. The reproducibility of atmospheric parameters in the WRF model deteriorated under stable thermal stratification of the atmosphere. The same issue was pointed out in [38], which was a fine-resolution WRF simulation of stably stratified flows in shallow pre-alpine valleys. The authors found that the diurnal temperature range was underestimated in the WRF model;

(ii)  The structure of turbulence over the BAO significantly depended on the orography and characteristics of meso-scale atmospheric disturbances (vortex and jet streams). The BAO is located at the periphery of a meso-scale atmospheric vortex structure with an anticyclonic direction of air flows in the daytime. An analysis of the energy spectra showed that the characteristic scales of this meso-scale vortex structure, whose center is formed north of the BAO, vary from 10 to 20 km;

(iii) We showed that the increase in image quality was due to weakening of the airflow over Lake Baikal, as well as a decrease in meso-scale wind speed fluctuations. In comparison with nighttime characteristics, the daytime spectral characteristics of the wind speed fluctuations, $E_n$ and $\int_{f_1}^{f_2} E(f)df/Int$, increased 2.5–3.3 and 2.0–3.4 times, respectively. The energy of high-frequency fluctuations in wind speed during the day also significantly increased.

(iv)  We showed that the daytime spectrum of atmospheric meso-scale turbulence was close to the classical spectrum of strong turbulence (described by the "−5/3" law). At night and in the morning, the spectrum has a steeper slope on small scales. The spatial scales, based on which the spectrum changes its slope from "−5/3" to ∼ "−3", are associated with the suppression of atmospheric turbulence in stably stratified atmospheric layers. The characteristic scales of the transition between different regions of the spectrum were 2–2.5 km.

The obtained results are relevant for the development of methods to predict optical turbulence for ground-based telescopes, as well as the assessment of local transport of impurities [39,40]. In the future, we plan to investigate the influence of baroclinic large-scale and meso-scale structures of atmospheric flow on dynamic and optical turbulence. These studies contribute to the development of methods for the diagnostics and prediction of astronomical image quality.

Furthermore, WRF simulation is known to be sensitive to the type of atmospheric event and the combinations of physical parametrizations [41]. We need to evaluate the possibilities of modeling meso-scale processes above the observatory, taking into account different atmospheric events and parameterizations, including schemes based on the turbulent kinetic energy and its temporal transformations. Moreover, an interesting area of further research would be to perform detailed studies above the Sayan Solar Observatory at a height of 2000 m above sea level. For SSO, we plan to consider atmospheric cases with high image quality.

**Author Contributions:** Investigation, visualization, writing—review and editing: A.Y.S. and P.G.K.; methodology: P.G.K., A.Y.S. and A.A.L.; formal analysis, investigation, visualization: A.V.K. and O.A.K.; writing—review and editing: I.V.R. and P.G.K.; software, visualization: A.V.K., D.Y.K. and M.Y.S. All authors have read and agreed to the published version of the manuscript.

**Funding:** This research was funded by the RSF grant No. 22-29-01137.

**Institutional Review Board Statement:** Not applicable.

**Informed Consent Statement:** Not applicable.

**Data Availability Statement:** Data used are available on request from the corresponding author.

**Acknowledgments:** Measurements were carried out using the unique research facility large solar vacuum telescope, http://ckp-rf.ru/usu/200615/ (accessed on 1 December 2022).

**Conflicts of Interest:** The authors declare no conflict of interest.

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
