# Peer review of "Influence of Atmospheric Flow Structure on Optical Turbulence Characteristics"

_applsci, doi:10.3390/app13031282_

Round 1
Reviewer 1 Report
The received manuscript concerns the quality of astronomical images under conditions of moderate small-scale turbulence and varying meso-scale airflows above the Baykal Astrophysical Observatory. The methods applied are a Weather Research and Forecasting (WRF) model as well as statistical estimations of the Fried parameter from the differential motion of the solar images.
The topic is relevant and make the study very interesting. The results obtained are important for the development of prediction of optical turbulence for ground-based telescopes, and the assessment of local transport of impurities which determines their practical application.
There is rather proper clarity of the text including the table and graph part. However, there is no broader discussion of the received results which I recommend supplementing. The manuscript is well summarized and clearly concluded.
Author Response
We sincerely thank the anonymous reviewer for his work and useful recommendations. We have added discussion section. Thank you very much.

Reviewer 2 Report
Manuscript Number: applsci-2140614
Full Title: Influence of atmospheric flow structure on optical turbulence characteristics
I – General Comments
The manuscript deals with a Weather Research and Forecasting (WRF) model as well as statistical estimations of the Fried parameter from the differential motion of the solar images. The authors’ purpose is to discuss the quality of astronomical images under conditions of moderate small-scale turbulence and varying meso-scale airflows above the Baykal Astrophysical Observatory (BAO). The obtained results have shown that the parameterization schemes used in the Weather Research and Forecasting model and the analysis of optical turbulence strength (namely, the Fried parameter) present a good agreement with the expected physics. There are important recommendations/questions addressed to the authors, before to accept their manuscript to be published as Applied Science’s paper.
II - Specific Comments
(i) In “Abstract”, the main contribution of the present work into the specialized literature context must be better clarified aiming to justify the intended publication.
(ii) In “Introduction”, the authors have clarified the key idea of their manuscript; that is to diagnose and predict the small-scale turbulence and the solar image quality. The authors have also mentioned their purpose of expanding knowledge about the structured turbulent small-scale motions and optical turbulence, in which comparisons between the measurements of solar image differential motion and Weather Research and Forecasting model data has been performed. However, the authors must complement the “Introduction” by considering: a) some more detailed information on the specific topic of the research into the Applied Sciences’ scope by introducing previous works; b) a description of the exact question or hypothesis that the paper will address; and c) a summary of the adopted approach to solve the investigated problem. Therefore, it is necessary to present more technical discussions concerning these key points including examples of past models through a critical review. As consequence, the contributions of the manuscript can better explained and contextualized in this part of the paper.
(iii) In Section 2, it is important to include some specific comment concerning the assumed hypothesis, governing equations and boundary conditions for the chosen problem. A figure 1 can be properly linked with the idea of the general formulation of the problem.
(iv) How is dissipated the energy from the macro scales to small ones? How to link the concept of turbulent viscosity into the present methodology?
(v) How can be established a velocity profile for atmospheric flows involving different layers?
(vi) The sensitivity of the results by comparing rough terrain and smoothed terrain must be better contextualized and interpreted. What is expected by comparing results with and without surface roughness effects?
(vii) How is affected the expected physics by adopting the isothermal hypothesis? The authors have commented that the reproducibility of atmospheric parameters in the WRF model deteriorates under stable thermal stratification of the atmosphere. Please, include more details about it.
(viii) Figures (7)-(10) present the same style. It is suggested (if possible) a comparison with other methodologies previously developed in the specialized literature. If no, please, include a justification about.
(ix) A discussion about wind turbulence intensity during the experiments is welcome. It is suggested to substitute “vortex” by “vortical structure” into the manuscript too.
(x) In “Conclusions”, it is necessary to include comments with respect the numerical results behavior as compared as previous works (specially, experimental data, when possible). In closing, it is important to complete the manuscript with perspectives for a future research. Finally, the main contribution of the present manuscript should be clarified aiming to justify its publication in Applied Sciences.
III - Recommendation for the Applied Sciences´ Editor
In my opinion, the present manuscript needs attend all topics above presented. Upon consideration of all points above, I think the paper could be considered for publication in Applied Sciences.
Author Response
We sincerely thank the anonymous reviewer for his work and foreseeable recommendations. Thank you very much. We have improved the manuscrpit according to your recommendations.
(i) In “Abstract”, the main contribution of the present work into the specialized literature context must be better clarified aiming to justify the intended publication.
We have added information in the abstract in lines 4 – 11.
(ii) In “Introduction”, the authors have clarified the key idea of their manuscript; that is to diagnose and predict the small-scale turbulence and the solar image quality. The authors have also mentioned their purpose of expanding knowledge about the structured turbulent small-scale motions and optical turbulence, in which comparisons between the measurements of solar image differential motion and Weather Research and Forecasting model data has been performed. However, the authors must complement the “Introduction” by considering: a) some more detailed information on the specific topic of the research into the Applied Sciences’ scope by introducing previous works; b) a description of the exact question or hypothesis that the paper will address; and c) a summary of the adopted approach to solve the investigated problem. Therefore, it is necessary to present more technical discussions concerning these key points including examples of past models through a critical review. As consequence, the contributions of the manuscript can better explained and contextualized in this part of the paper.
We added information and references. The major question of the present study is to identify possible ways in which large-scale and meso-scale atmospheric structures influence small-scale turbulence and image quality within the observatory region. Also, in order to diagnose and predict the small-scale turbulence and the solar image quality, the present study is aimed at expanding knowledge about the structured turbulent small-scale motions and optical turbulence. We would like also to emphasize that rather little data is contained in the literature on the study of processes in rough terrain near large cold water bodies. Our research is the first for the selected region.
(iii) In Section 2, it is important to include some specific comment concerning the assumed hypothesis, governing equations and boundary conditions for the chosen problem. A figure 1 can be properly linked with the idea of the general formulation of the problem.
We add comments concerning the assumed problem and the main governing equations.
(iv) How is dissipated the energy from the macro scales to small ones? How to link the concept of turbulent viscosity into the present methodology?
We add equations 1-3 used in the study
(v) How can be established a velocity profile for atmospheric flows involving different layers?
We add velocity profiles for atmospheric flow
(vi) The sensitivity of the results by comparing rough terrain and smoothed terrain must be better contextualized and interpreted. What is expected by comparing results with and without surface roughness effects?
In calculations we used model of relief. We add discussion about this problem.
(vii) How is affected the expected physics by adopting the isothermal hypothesis? The authors have commented that the reproducibility of atmospheric parameters in the WRF model deteriorates under stable thermal stratification of the atmosphere. Please, include more details about it.
We add discussion section
(viii) Figures (7)-(10) present the same style. It is suggested (if possible) a comparison with other methodologies previously developed in the specialized literature. If no, please, include a justification about.
We used standart python packages for wrf coordinate transformations as well as interpolation models.
(ix) A discussion about wind turbulence intensity during the experiments is welcome. It is suggested to substitute “vortex” by “vortical structure” into the manuscript too.
We add discussion section
(x) In “Conclusions”, it is necessary to include comments with respect the numerical results behavior as compared as previous works (specially, experimental data, when possible). In closing, it is important to complete the manuscript with perspectives for a future research. Finally, the main contribution of the present manuscript should be clarified aiming to justify its publication in Applied Sciences.
We add the required information.

Round 2
Reviewer 2 Report
Manuscript Number: applsci-2140614-v2
Full Title: Influence of atmospheric flow structure on optical turbulence characteristics
The original manuscript has been satisfactory revised by authors. In my opinion, the present manuscript can be published as Applied Science’s paper.